# Multilingual Knowledge Graph Completion
# With Joint Relation and Entity Alignment

**Harkanwar Singh**                                      HARKANWARSINGH1841@GMAIL.COM
*Indian Institute of Technology Delhi*

**Soumen Chakrabarti**                                      SOUMEN@IITB.AC.IN
*Indian Institute of Technology Bombay*

**Prachi Jain**                                      P6.JAIN@GMAIL.COM
*Indian Institute of Technology Delhi*

**Sharod Roy Choudhury**                                      SHAROD.ROY@GMAIL.COM
*Indian Institute of Technology Bombay*

**Mausam**                                      MAUSAM@CSE.IITD.AC.IN
*Indian Institute of Technology Delhi*

## Abstract

Knowledge Graph Completion (KGC) predicts missing facts in an incomplete Knowledge Graph. We study *Multilingual* KGC for KGs that associate entities and relations with surface forms from different languages. In such a setting, an entity (or relation) may be mentioned as separate IDs in different KGs, necessitating entity alignment (EA) and relation alignment (RA). In addition to EA and RA being important subtasks for Multilingual KGC, we posit that high confidence fact predictions may also, in turn, add valuable information for alignment tasks, and vice versa. In response, we present the novel task of jointly training multilingual KGC, EA and RA models. Our approach, ALIGNKGC, uses mBERT-based surface form overlap for EA, and combines it with a KGC approach, extended to the multiple KG setting via a loss term that incentivizes RA. On experiments with DBPedia in five languages, we find that ALIGNKGC achieves up to 17% absolute MRR improvements in KGC compared to a strong completion model that combines known facts in all languages. It also outperforms an mBERT-only alignment baseline for EA, underscoring the value of joint training for these tasks.

## 1. Introduction

A knowledge graph (KG) has nodes representing entities and directed edges representing relations between subject and object entities. An entity has a unique node (ID). Relations also have canonical labels such as born-in or works-at, with associated IDs. A KG is usually associated with a single human language — an entity node (or relation ID) is associated with one or more surface forms in that language. E.g., the ID for the country USA may have aliases like "United States of America".

KGs are usually very incomplete, as curators struggle to keep up with the real world. KG completion (KGC) is thus a strongly motivated problem and studies the prediction of true facts unknown to the KG. While the problem has been intensively researched over the last few years [Bordes et al., 2013, Trouillon et al., 2016, Jain et al., 2018, Sun et al., 2019], most KGC research is applicable to only one KG in one language at a time. However, different language speakers would maintain separate KGs in their own languages. Independent completions of each KG may not be optimal, since information from one KG will likely help completion of the other.

Each KG will give a different ID, and often also a different surface form, to the same entity, such as "Estados Unidos de América" for the USA entity in a Spanish KG. This leads to the problem of ID proliferation. Recent work on entity alignment (EA) across KGs attempts to assign a unique ID to all nodes representing the same entity [Chen et al., 2017, Sun et al., 2017, 2018, 2020, Cao et al., 2019, Chen et al., 2021, Tang et al., 2020]. A related but under-explored task is relation alignment (RA) — assigning synonymous relations in different KGs the same relation ID. We note that RA involves *global* evidence, because the decision to merge two relations in two languages can have far-reaching consequences to many facts in both KGs.

Our key contribution is to recognize and exploit the synergy between multilingual KGC, EA and RA tasks. Entity and relation alignments expose a KGC algorithm to more facts, which can lead to better completion. Conversely, a high-confidence completion can give additional evidence to align its constituent entities and relation. (See Appendix A for an extended example.) In this paper, we present ALIGNKGC, a multi-task system that learns to optimize for KGC, EA and RA jointly. ALIGNKGC uses state-of-the-art mBERT-based models for EA and combines it with an extension of the state-of-the-art ComplEx model [Trouillon et al., 2016] for single KGC. ALIGNKGC uses a novel subject-object signature of each relation, which it represents as a bag of embeddings, and compares them for equivalence and implication between relations. ALIGNKGC uses this information as additional term in its loss function for better RA and multilingual KGC performance.

We evaluate ALIGNKGC on slices of DBPedia in five languages. We compare it against two single-KGC systems, one trained over each monolingual KG separately, and the other trained over a union of all KG facts. Compared to these strong baselines, we find that ALIGNKGC achieves substantial accuracy boost due to better entity and relation alignment, obtaining up to 17% absolute MRR improvements in KGC across languages, around 20% absolute HITS@1 gain for RA for rare relations, around 26% absolute HITS@1 gain for RA for more frequent relations, and around 61% absolute HITS@1 gain for EA. ALIGNKGC also achieves a 5.3% absolute HITS@1 gain over an even stronger EA baseline that uses mBERT embeddings on entity names. These results underscore the value of joint training for these tasks.

## 2. Notation and preliminaries

A knowledge graph (KG) provides a graph-view to a knowledge-base (KB). A KG consists of entities $E$ and relations (aka relation types) $R$. A KG instance is a triple $(s, r, o)$ where $s, o \in E$ and $r \in R$. These are all canonical IDs. Each ID is associated with one or more textual aliases.

**KGC task:**  For any single KG, training data is provided as $(s, r, o)$ triples. A test instance has the form $(s, r, ?)$ or $(?, r, o)$ where the system has to predict $o$ or $s$. Multiple correct values are possible. The evaluation protocol takes a system-ranked list of candidate objects or subjects, and then measures MRR or hits@$K$. Many KG embedding and KGC algorithms have been proposed in the last few years. ComplEx [Trouillon et al., 2016] with all negative instances (no sampling) is a simple model that gives near state-of-the-art predictions [Jain et al., 2020a]. It has also formed the basis for many KGC extensions (e.g., [Jain et al., 2020b]). So we use it as our baseline KGC gadget. ComplEx defines a triple score as $f(s, r, o) = \Re\left(\langle \boldsymbol{s}, \boldsymbol{r}, \boldsymbol{o}^{\star} \rangle\right)$, where $c^{\star}$ is complex conjugate, $\langle \cdots \rangle$ is a 3-way elementwise inner product and $\Re(\cdot)$ is the real part of a complex number. Using $f$, ComplEx defines

$$\Pr(o|s, r) = e^{f(s,r,o)} \Big/ \sum_{o'} e^{f(s,r,o')}, \qquad \Pr(s|o, r) = e^{f(s,r,o)} \Big/ \sum_{s'} e^{f(s',r,o)} \qquad (1)$$

and the log-likelihood **KGC loss** as $L_{\text{KGC}} = \sum_{(s,r,o) \in \text{KG}} -\log \Pr(o|s, r) - \log \Pr(s|o, r)$.

**Alignment tasks:** We consider a set $L = \{l_1, l_2, \ldots\}$ of languages. $KG_l$ represents the KG supported by the language $l$. An entity in this KG is called $e_l$. A relation in this KG is called $r_l$. While trying to improve KGC for target language $l'$, KGs in multiple source languages $l$ may be combined.

An equivalence between entities $e_l$ and $e_{l'}$ in KGs of two languages $l$ and $l'$ is specified as $e_l \equiv e_{l'}$, also written as the triple $(e_l, \equiv, e_{l'})$. Similarly, an equivalence between relations $r_l$ and $r_{l'}$ is specified as $r_l \equiv r_{l'}$. Other types of alignment may be possible between relation pairs, such as $r_l \Rightarrow r_{l'}$, which means, for all $s, o$ such that $(s, r_l, o)$ holds, so does $(s, r_{l'}, o)$. During training, a set of entity equivalences $\{(e_l^n, \equiv, e_{l'}^n) : n = 1, \ldots, N\}$ and a set of relation equivalences $\{(r_l^m, \equiv, r_{l'}^m) : m = 1, \ldots, M\}$ are revealed to the system between each $KG_l$ and $KG_{l'}$. The goal of the system is to infer additional entity and relation equivalences. The system is usually called upon to produce a ranked list of equivalences, which is evaluated using HITS@$K$ or MRR.

To achieve KGC enhanced with alignment, the system has to infer additional triples in each KG, making best use of the revealed equivalences, while also inferring additional alignment triples between entities or relations across KGs of different languages.

## 3. Proposed methods

### 3.1 KGC baselines

When ComplEx is applied to any one KG in isolation, we call this method **KGCMONO**. Perhaps the most straight-forward way to apply a KGC system across multiple KGs is to compute $KG_U = \bigcup_{l \in L} KG_l$ such that for all revealed entity equivalences $\{(e_l, \equiv, e_{l'})\}$, collapse their node pairs, and rename all equivalent $r_l, r_{l'}$ relation IDs to a common new ID. A standard KGC system can work on the resulting $KG_U$, and we call this method **KGCUNION**.

### 3.2 Joint alignment and completion

For both textual and structured inputs, the problem of inferring a predicate or relation as entailed by another has been well-studied [Lin and Pantel, 2001, Bhagat et al., 2007, Nakashole et al., 2012]. With that work as our point of departure, we ask: How similar are two relations $r_l, r_{l'}$ in $KG_U$? Comparing their embedding vectors $\boldsymbol{r}_l, \boldsymbol{r}_{l'}$, as computed by a KGC algorithm such as ComplEx, gives an estimate. However, when training these embeddings, we found that a simple union of facts (KGCMONO) did not yield the best results. Instead, guiding the training through an auxiliary loss, computed using the following explicit *relation signatures* helped the model learn better embeddings. Here $s, o$ are interpreted as canonical IDs (hence, 'hard'). This will be generalized shortly.

**Definition 1** (**Hard SO-signature**). The ('hard') subject-object (SO) signature of a relation is defined as $\mathrm{SO}(r) = \{(s, o) : (s, r, o) \in KG_U\}$.

3.2.1 HARD SO-OVERLAP & JACCARD SIMILARITY

**Definition 2** (**SO-overlap**). The (hard) overlap between two relations $r_l, r_{l'}$ is $|\mathrm{SO}(r_l) \cap \mathrm{SO}(r_{l'})|$.

Jaccard similarity can then be used as a standard symmetric belief that two relations IDs in different languages are equivalent:

$$b_J(r_l \Leftrightarrow r_{l'}) = \frac{|\mathrm{SO}(r_l) \cap \mathrm{SO}(r_{l'})|}{|\mathrm{SO}(r_l) \cup \mathrm{SO}(r_{l'})|}. \tag{2}$$

If $b_J(r_l \Leftrightarrow r_{l'})$ exceeds a threshold $\theta$ (tuned using dev set), we add $(r_l, r_{l'})$ to set $\mathcal{A}_J$ of 'silver' alignments. We define an additional RA loss term encouraging such embeddings to be aligned:

$$L_{\text{RA-J}} = \sum_{(r_l, r_{l'}) \in \mathcal{A}_J} b_J(r_l \Leftrightarrow r_{l'}) \|\boldsymbol{r}_l - \boldsymbol{r}_{l'}\|_1. \tag{3}$$

We call this scheme **Jaccard**.

### 3.2.2 ASYMMETRIC SUBSUMPTION

A problem with Jaccard similarity is that it can give a symmetric high score to relation pairs having asymmetric implications between them. E.g., in the DBP5L data set, Jaccard similarity gives a large symmetric similarity score between locationCity and headquarter, or keyPerson and founders relations in different KGs. However, these relation pairs clearly involve asymmetric implication, and should not be linked.

Therefore, we need an asymmetric belief measure ($b_A$) for one relation subsuming another, which we define as

$$b_A(r_l {\Rightarrow} r_{l'}) = \frac{|\operatorname{SO}(r_l) \cap \operatorname{SO}(r_{l'})|}{|\operatorname{SO}(r_l)|} \in [0, 1] \tag{4}$$

The asymmetry in (4) refrains from pushing those relation pairs together. This avoids some erroneous inferences, generally boosting precision — possibly at the expense of some recall, as this may also remove some "reasonable inferences". Extending the logical statement $(r_l \Leftrightarrow r_{l'})$ iff $(r_l {\Rightarrow} r_{l'}) \wedge (r_{l'} {\Rightarrow} r_l)$ to the fuzzy domain, we get $b_A(r_l {\Leftrightarrow} r_{l'}) = \min\{b_A(r_l {\Rightarrow} r_{l'}), b_A(r_{l'} {\Rightarrow} r_l)\}$, which we readily see as equivalent to

$$|\operatorname{SO}(r_l) \cap \operatorname{SO}(r_{l'})| \min\left\{ \tfrac{1}{|\operatorname{SO}(r_l)|}, \tfrac{1}{|\operatorname{SO}(r_{l'})|} \right\} = \tfrac{|\operatorname{SO}(r_l) \cap \operatorname{SO}(r_{l'})|}{\max\{|\operatorname{SO}(r_l)|, |\operatorname{SO}(r_{l'})|\}} \in [0, 1]. \tag{5}$$

Similar to Eqn. (3), we incentivize the model to align $r_l$ and $r_{l'}$ if their $b_A(r_l {\Leftrightarrow} r_{l'})$ is high. In particular, we search for pairs of 'silver' alignments $(r_l, r_{l'}) \in \mathcal{A}_A$, where $r_l = \operatorname{argmax}_{r'_l} b_A(r_{l'} \Rightarrow r'_l)$ and $r_{l'} = \operatorname{argmax}_{r'_{l'}} b_A(r_l \Rightarrow r'_{l'})$. In other words, we want to apply the loss only to partner relations with mutually largest implication beliefs in that language. We define an asymmetric RA loss as:

$$L_{\text{RA-A}} = \sum_{(r_l, r_{l'}) \in \mathcal{A}_A} b_A(r_l \Leftrightarrow r_{l'}) \|\boldsymbol{r}_l - \boldsymbol{r}_{l'}\|_1. \tag{6}$$

We call this method **Asymmetric**.

### 3.2.3 SOFT OVERLAP AND APPROXIMATION

In the above definitions involving SO, we assumed the $(s, o)$ pairs were represented using canonical entity IDs. Unless explicit entity equivalences are provided, 'hard' SO overlap will underestimate relation similarities. We now extend these ideas to redefine SO-signatures using entity embeddings, which can be trained via gradient descent.

Recall that the KGC system obtains embedding vectors $\boldsymbol{e}$ for each entity $e$ in the KG. Overloading notation, we modify our earlier definition of subject-object signature.

**Definition 3 (Soft SO signature).** For each relation $r$, $\operatorname{SO}(r) = \{(\boldsymbol{s}, \boldsymbol{o}) : (s, r, o) \in \operatorname{KG}_U\}$.

Each element is the concatenation of the subject and object *embedding vectors*. The $i$th embedding pair in $\operatorname{SO}(r)$ is denoted $\operatorname{SO}(r)[i]$. Two such pairs can be compared by, say, extending

cosine similarity with an AND-semantic: $\text{sim}\big((\boldsymbol{s},\boldsymbol{o}),(\boldsymbol{s}',\boldsymbol{o}')\big) = \sigma(\cos(\boldsymbol{s},\boldsymbol{s}'))\sigma(\cos(\boldsymbol{o},\boldsymbol{o}'))$, where $\sigma$ is the sigmoid nonlinearity. This captures the requirement that the subjects have to be similar *and* the objects have to be similar. The key challenge is to compare two *sets* of such SO vectors, $\text{SO}(r_l), \text{SO}(r_{l'})$ for extending Defn. 2. A simple approach is to use a network to encode each SO-set [Zaheer et al., 2017, Lee et al., 2019] into a set embedding and then compare these (see Appendix B for details). However, recent work [Pabbaraju and Jain, 2019, Tang et al., 2020] suggests that we should drill down to pairwise interactions between set elements. To that end, consider two relations from different languages, $r_l, r_{l'}$, that are candidates for alignment. Suppose the $(\boldsymbol{s},\boldsymbol{o})$ pairs of $r_l$ are indexed by $i$ and pairs of $r_{l'}$ are indexed by $j$. We build a matrix $A_{r_l,r_{l'}}$ of pairwise cosine similarities: $A_{r_l,r_{l'}}[i,j] = \text{sim}\big(\text{SO}(r_l)[i], \text{SO}(r_{l'})[j]\big)$.

**Definition 4** (**Soft SO-Overlap**). The continuous extension of $|\,\text{SO}(r_l) \cap \text{SO}(r_{l'})|$, denoted by $\text{SoftOv}(r_l, r_{l'})$, is defined as the value of the maximal matching on the weighted bipartite graph induced by $A_{r_l,r_{l'}}$.

Note that $\text{SoftOv}(r_l, r_{l'})$ depends on entity embeddings, which are to be trained. Ideally, we should be able to backpropagate various KGC and alignment losses past the solution of the matching problem to the entity and relation embeddings. Gumbel-Sinkhorn matrix scaling [Cuturi, 2013, Mena et al., 2018] can be used for this purpose, but it is computationally expensive at KG scales. (See Appendix C for details.) Here we use a computationally cheaper approximation: only if $i$ is $j$'s strongest partner ($i = \text{argmax}_{i'} A[i', j]$) and $j$ is $i$'s strongest partner ($j = \text{argmax}_{j'} A[i, j']$), we choose edge $(i, j)$ in our matching. For each such edge $(i, j)$, we add in $\text{SoftOv}(r_l, r_{l'})$ a score increment $\sigma\big(A_{r_l,r_{l'}}[i,j]\,w + c\big)$, where $\sigma$ is the sigmoid nonlinearity, and $w > 0, c \in \mathbb{R}$ are model parameters trained along with all embeddings.

We continue to use (4)-(5) as defined for **Asymmetric**, except we replace the hard overlap $|\,\text{SO}(r_l) \cap \text{SO}(r_{l'})|$ by $\text{SoftOv}(r_l, r_{l'})$. We denote the beliefs computed thus by $b_{SA}$ (for soft asymmetric). The rest of the machinery of the previous section follows, i.e., we compute 'silver' alignments $(r_l, r_{l'}) \in \mathcal{A}_{SA}$, leading to a soft asymmetric RA loss as:

$$L_{\text{RA-SA}} = \sum_{(r_l,r_{l'}) \in \mathcal{A}_{SA}} b_{SA}(r_l \Leftrightarrow r_{l'})\big\|\boldsymbol{r}_l - \boldsymbol{r}_{l'}\big\|_1. \tag{7}$$

We call the resulting method **SoftAsymmetric**.

### 3.2.4 ADDING TEXT SIGNALS TO ALIGNKGC

BERT-INT [Tang et al., 2020] demonstrated the power of using mBERT [Devlin et al., 2019] to derive text-based features that effectively pull together duplicate entities. A short text passage $t(e)$ is collected from the name (alias) and description of each entity $e$, which is then sent into mBERT, and the `[CLS]` embedding output by mBERT used as the textual representation of $e$. Optionally, to better map textual embeddings into the KG space, we use a small feed-forward network $\text{mlp}(\text{mbert}(t(e)))$, which, for economy of notation, we will write as $\text{mbert}(e)$. For entity alignment, we define an additional loss term $L_{\text{EA}} = \sum_{e_l,e_{l'}} \cos(\text{mbert}(e_l), \text{mbert}(e_{l'}))\|\boldsymbol{e_l} - \boldsymbol{e_{l'}}\|_1$ The intuition is that when the mBERT embeddings are similar, we want the $L_1$ distance between KG embeddings to be small, and vice versa. Because we found Asymmetric generally better than Jaccard, we add the above mBERT-based loss term $L_{\text{EA}}$ to two previous variations. When we add it to Asymmetric, we call the resulting system **Asym.+mBERT**. When we add it to SoftAsymmetric, the resulting system is called **SoftAsym.+mBERT**, which is the full form of **ALIGNKGC**, with all other variations regarded as

ablations. In a similar way, we can also obtain a textual embedding of a relation and use that to create a textual RA loss. In our test bed, we did not have reliable surface forms of relations in multiple languages, so we turned off this component. Further details of this loss are described in Appendix D.

## 4. Experiments

We study the following research questions through our experiments. (1) Does our joint KGC, EA and RA model (ALIGNKGC) improve upon independent models for each of these tasks? (2) Is there any cross-lingual transfer of knowledge, for facts that may be mentioned in one language, but are queried in another? (3) What are the incremental contributions of soft overlap and mBERT's surface form encoding in the overall performance of ALIGNKGC? (4) How does performance vary with the number of seed alignments (both entity and relation) provided at training time?

### 4.1 Experimental Settings

**Dataset:** Standard mono-lingual KGC datasets such as FB15k, FB15k-237, WN18, WN18RR, and YAGO-3-10, are not suited for evaluation of multi-lingual alignment or KGC. A few recent EA datasets such as DBP15K [Sun et al., 2017] and DB2.0 [Sun et al., 2021] provide gold EAs (or 'dangling' non-EAs in DB2.0) between *two* languages at a time. They do not provide direct support for synergistic training and evaluation of

| Language | Greek | Japanese | Spanish | French | English |
|---|---|---|---|---|---|
| **#Entity** | 5,231 | 11,805 | 12,382 | 13,176 | 13,996 |
| **#Relation** | 111 | 128 | 144 | 178 | 831 |
| **#Triples** | 13,839 | 28,774 | 54,066 | 49,015 | 80,167 |

Table 1: Salient statistics of the DBP5L data set.

RA and KGC at the same time. Adaptations of such benchmarks in either direction appear fraught with potential unfairness. If we evaluate ALIGNKGC on two languages at a time, it loses its essential advantage of EAs across KGs in *multiple* languages, together with integrated RA and KGC supervision. On the other hand, if we collapse all but the target language into a "super language", we severely distort the distribution of entity equivalence class sizes and frequency of relations, for other systems.

Most suited for our purpose is the recently-released DBP5L benchmark [Chen et al., 2020], which is derived from DBPedia in five languages: English (En), Greek (El), Spanish (Es), Japanese (Ja) and French (Fr). We use the same 60-30-10 splits of the KG triples into train-dev-test folds as in Chen et al. [2020] and combine the train sets of all source languages for training. Importantly, DBP5L also provides about 40% EAs between language pairs. We randomly pick half of seed entity alignments for training; the rest are used for testing. Because DBPedia uses a uniform relation vocabulary that is normalized across all languages, it cannot be directly

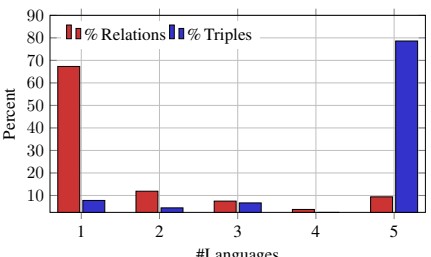

Figure 2: Fraction of relation labels and fact tuples with the number of languages they appear in.

used to test relation alignment. To test RA capabilities of our models, we create separate ID spaces for each language. The original normalized IDs create gold data for alignment. The models do not get any seed relation alignments, and the whole alignment data is used for testing RA. Note that no asymmetric gold relation implication $r_1 \Rightarrow r_2$ are available, only equivalences $r_1 \equiv r_2$.

| | GREEK | | | ENGLISH | | | SPANISH | | | FRENCH | | | JAPANESE | | |
|---|---|---|---|---|---|---|---|---|---|---|---|---|---|---|---|
| **Models** | H@1 | H@10 | MRR | H@1 | H@10 | MRR | H@1 | H@10 | MRR | H@1 | H@10 | MRR | H@1 | H@10 | MRR |
| KGCMONO | 23.6 | 49.0 | 31.9 | 18.8 | 43.0 | 26.9 | 22.1 | 50.1 | 31.4 | 24.0 | 50.4 | 32.8 | 26.4 | 49.4 | 34.4 |
| KEnS$_b$(RotatE) | 27.5 | 56.5 | - | 14.4 | 39.6 | - | 25.2 | 62.6 | - | 22.3 | 60.6 | - | 32.9 | 64.8 | - |
| KGCUNION | 39.0 | 75.9 | 52.1 | 25.3 | 51.7 | 34.3 | 34.9 | 66.3 | 45.6 | 37.5 | 70.3 | 48.5 | 40.3 | 69.9 | 50.9 |
| Jaccard | 49.8 | 82.2 | 60.9 | 26.5 | 53.5 | 35.6 | 39.2 | 69.4 | 49.6 | 40.9 | 71.6 | 51.6 | 45.8 | 73.6 | 55.6 |
| Asymmetric | 51.0 | 82.9 | 62.4 | 26.3 | 53.2 | 35.3 | 40.4 | 69.4 | 50.3 | 40.5 | 72.4 | 51.6 | 43.2 | 73.2 | 54.1 |
| Asym.+mBERT | 52.8 | 85.3 | 64.7 | 29.1 | 57.3 | 38.6 | 42.9 | 73.6 | 53.6 | 44.2 | 76.7 | 55.6 | 46.3 | 76.3 | 57.3 |
| SoftAsymmetric | 55.1 | 84.2 | 65.5 | 28.5 | 54.9 | 37.5 | 44.0 | 71.4 | 53.4 | 44.5 | 74.0 | 54.9 | 46.7 | 74.4 | 56.6 |
| SoftAsym.+mBERT | **58.2** | **88.6** | **69.4** | **31.7** | **59.8** | **41.3** | **48.0** | **76.6** | **58.0** | **48.4** | **79.4** | **59.5** | **49.3** | **78.7** | **60.1** |
| **UNSEEN TEST SET** | | | | | | | | | | | | | | | |
| KGCUNION | 25.4 | 64.3 | 38.5 | 19.4 | 45.6 | 28.1 | 22.7 | 55.2 | 33.5 | 27.3 | 61.1 | 38.3 | 27.0 | 57.7 | 37.7 |
| Jaccard | 28.3 | 69.9 | 41.7 | 19.7 | 47.1 | 28.8 | 24.9 | 58.5 | 36.0 | 28.2 | 61.6 | 39.5 | 29.4 | 61.3 | 40.1 |
| Asymmetric | 28.3 | 70.0 | 42.2 | 18.9 | 46.8 | 28.0 | 25.0 | 58.2 | 35.9 | 26.7 | 62.1 | 38.6 | 26.1 | 60.6 | 38.0 |
| Asym.+mBERT | 33.2 | 75.1 | 47.8 | 21.9 | 51.3 | 31.6 | 28.3 | 64.1 | 40.3 | 30.9 | 68.6 | 43.7 | 30.5 | 65.2 | 42.8 |
| SoftAsymmetric | 32.3 | 72.4 | 45.8 | 20.0 | 48.4 | 29.5 | 26.7 | 60.8 | 38.0 | 29.2 | 39.6 | 41.1 | 27.3 | 61.9 | 39.3 |
| SoftAsym.+mBERT | **37.2** | **80.0** | **52.1** | **23.5** | **54.0** | **33.6** | **32.0** | **67.8** | **44.2** | **34.1** | **71.7** | **47.2** | **31.6** | **68.3** | **44.5** |
| **SEEN TEST SET** | | | | | | | | | | | | | | | |
| KGCUNION | 56.1 | 90.5 | 69.2 | 65.8 | 93.5 | 76.3 | 66.6 | 95.3 | 77.2 | 64.3 | 94.4 | 75.3 | 66.3 | 93.7 | 76.8 |
| Jaccard | 76.9 | 97.8 | 85.0 | 73.1 | 97.2 | 82.4 | 76.7 | 98.0 | 85.1 | 64.9 | 97.8 | 83.3 | 77.9 | 97.7 | 85.7 |
| Asymmetric | 78.3 | 98.3 | 86.5 | 76.8 | 98.1 | 85.3 | 79.5 | 98.6 | 87.1 | 76.7 | 98.2 | 85.4 | 76.8 | 97.7 | 85.5 |
| Asym.+mBERT | 77.4 | 98.1 | 85.8 | 78.4 | 98.3 | 86.4 | 81.2 | 98.6 | 88.2 | 79.1 | 97.9 | 86.7 | 77.2 | 98.1 | 85.8 |
| SoftAsymmetric | 83.7 | 99.1 | 90.1 | 86.8 | 99.6 | 92.5 | 89.1 | 99.3 | 93.6 | 84.6 | 99.0 | 90.9 | **84.6** | 98.8 | **90.6** |
| SoftAsym.+mBERT | **84.6** | **99.4** | **91.1** | **88.3** | **99.7** | **93.4** | **89.8** | **99.6** | **94.1** | **85.9** | **99.5** | **91.8** | 83.9 | **99.0** | **90.6** |

Table 3: KGC performance on five languages. SoftAsym.+mBERT is the full form of ALIGNKGC.

Table 1 lists the statistics of the five KGs in DBP5L. Unsurprisingly, En is the most well-populated, with 5–7 times more relations than other KGs. Figure 2 shows that a majority of the relation labels have associated string descriptions in only one of five languages (usually English). However, relations with monolingual aliases account for only 8% of fact triples. Meanwhile, almost 80% of fact triples use a relation label that has aliases in all five languages.

**Performance metrics:** We evaluate ALIGNKGC on all the three tasks — KGC, EA and RA. As is common when evaluating a system for KGC, we regard test instances $(s, r, ?)$ as a task of ranking $o$ (on the basis of scores computed in equations (1)), with gold $o^*$ known. We report MRR (Mean Reciprocal Rank) and the fraction of queries where $o^*$ is recalled within rank 1 and rank 10 (HITS). The *filtered* evaluation removes valid train or test tuples ranking above $(s, r, o^*)$ for scoring purposes.

To evaluate the system for EA, we use the test instance $(e_l, \equiv, ?_{l'})$ as a task of ranking $e_{l'}$s, using the cosine distance between the entity embeddings of the language pair. We calculate Hits@1 and Hits@10 on the resulting rankings. A similar evaluation is done for RA by ranking $r_{l'}$s for the query $(r_l, \equiv, ?_{l'})$. All pairs of languages are considered as $l$ and $l'$ for these evaluations.

**Algorithms compared:** For KGC, our baselines are KGCMONO and KGCUNION, and the recent multilingual KGC method called Knowledge Ensemble (KEnS) [Chen et al., 2020]. KEnS had strictly more information than ALIGNKGC, since it had access to aligned relation IDs at train time. We report scores from its best reported setting: KEnS$_b$(RotatE).

The full form of ALIGNKGC (SoftAsym.+mBERT) uses the following loss function:

$$L_{\text{KGC}} + \alpha L_{\text{reg}} + \beta L_{\text{RA-SA}} + \gamma L_{\text{EA}}. \tag{8}$$

Here $L_{\text{reg}}$ is an $L_2$ regularization on embeddings. $\alpha, \beta, \gamma \geq 0$ are tuned hyperparameters. We also evaluate other variations described in Section 3.2.

To understand the marginal utility of joint training over pre-trained knowledge in EA, we also compare against an extremely competitive mBERT-only baseline, since it is known that mBERT, by itself, can effectively align entities, given their aliases in different languages [Tang et al., 2020].

| Models | Relation Alignment(<500) | | Relation Alignment($\geq$500) | | Entity Alignment | |
|---|---|---|---|---|---|---|
| | Hits@1 | Hits@3 | Hits@1 | Hits@3 | Hits@1 | Hits@10 |
| KGCUNION | 19.8 | 28.7 | 42.8 | 66.5 | 23.5 | 42.8 |
| Jaccard | 28.6 | 40.1 | 62.4 | 75.6 | 37.5 | 55.4 |
| Asymmetric | 30.0 | 41.5 | 68.6 | 77.5 | 39.0 | 55.5 |
| Asymmetric+mBert | 31.7 | 44.9 | **70.0** | 77.0 | 82.2 | 90.0 |
| SoftAsymmetric | 29.5 | 40.0 | 65.9 | 75.1 | 50.2 | 65.4 |
| mBERT | - | - | - | - | 79.5 | 90.6 |
| SoftAsym.+mBERT | **39.1** | **50.9** | 69.1 | **78.5** | **84.8** | **91.9** |

Table 4: Entity and relation alignment performance of various models on five languages. Relations occurring in <500 and $\geq$500 triples are evaluated separately. (DBP5L does not provide usable relation description texts in multiple languages, so mBERT cannot be used for RA.)

Since completion is most useful around nascent entities, we use the entity surface form(s) as $t(e)$ and not, e.g., its full Wikipedia description or infoboxes.

**Training policies and hyperparameters:** We fine-tune mBERT on entity surface forms to get 300-dim vectors. We use Adagrad as the optimizer and fine tune hyper-parameters on a dev set. We select among the following sets of hyper-parameter values: learning rate from $\{0.2, 0.4, 0.6, 0.8, 1\}$, $\alpha$ from $\{0.002, 0.02, 0.2\}$, $\beta$ from $\{1, 2, 5, 10\}$, and $\gamma$ from $\{1, 5, 10, 50\}$. We fix a batch size of 500. Our best choices for ALIGNKGC were: learning rate=0.8, $\alpha$=0.02, $\beta$=5, and $\gamma$=50. We choose $\theta$=0.01 for Jaccard. We use a negative sampling of 2000 instances for each positive triple when training ComplEx.

We sample initial entity and relation embeddings via $\mathcal{N}(0, 0.05)$. As a form of curriculum, we let relation alignments stabilize over a few iterations and then make the equivalence scores trainable.

### 4.2 Results: KGC performance

Table 3 reports KGC performance on the five languages. Not surprisingly, all models outperform KGCMONO, since it does not perform multilingual training. KGCUNION obtains much better scores because of combining different language KGs. Both baselines are outperformed by all ALIGNKGC variants. Each successive model enhancement — Jaccard, Asymmetric Model, Soft Asymmetric Model, and addition of mBERT — achieves progressively better performance. mBERT also provides similar boost to the 'hard' Asymmetric variant, but Asym.+mBERT is beaten by SoftAsym.+mBERT.

To analyze the performance further, we split the test set into two subsets: seen and unseen. The *seen* test set refers to those test facts that are already seen at train time, but in a KG of a different language. I.e., a test fact $(s_l, r_l, o_l)$ is in seen test if $\exists (s_{l'}, r_{l'}, o_{l'}) \in \text{KG}_{l'}(train)$ s.t. $s_l \equiv s_{l'}$, $o_l \equiv o_{l'}$ and $r_l \equiv r_{l'}$, even though the model may/may not know these alignments. All other test facts are part of the *unseen* test set. Qualitatively, performance on the seen split represents the capacity of the model to memorize known facts in a language and align them to another language; the unseen split gives a true sense of a system's inference capability, since the fact has likely not been read in *any* language at train time (except if an entity alignment is missing).

We first note that the performance on seen test is generally high, suggesting that the models do a good job of memorizing and aligning facts. We find that Jaccard signficantly helps with seen split, since it is able to learn similar embeddings for two aligned relations, and yields high scores for seen tuple in a different language. Asymmetric loss helps further as it removes false positives in RA, suffered by (symmetric) Jaccard. But it has a slight negative effect on unseen test instances, likely because of removal of some reasonable inferences. Making SO-overlap soft and trainable

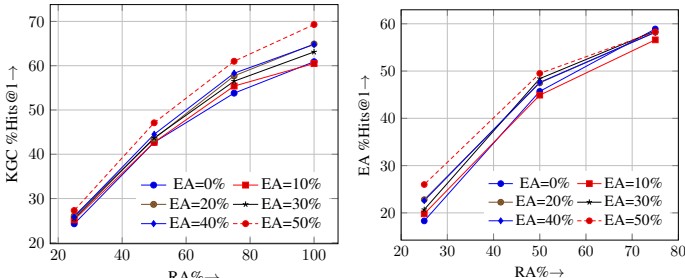

Figure 5: (a) KGC and (b) EA %Hits@1 as training %age of EA and RA are jointly varied.

enables more information flow, and better learning of embeddings and implication scores. This leads to significant improvement in both test splits. Finally, addition of mBERT yields massively better entity alignments, which has an impact on all tasks.

We notice that KEnS performs even worse than KGCUNION, even though it has knowledge of all relation alignments. We attribute this to (1) a stronger KGC baseline of ComplEx (with high negative sampling), compared to KEnS's use of RotatE with only one negative sample, and (2) ALIGNKGC's hard alignment policy of providing aligned entities in one unified KG with a single embedding, instead of KEnS's approach of finding nearest entities in each language KG separately and then creating an ensemble.

### 4.3 Results: Alignment performance

Table 4 reports the performance of the various models on RA and EA tasks. We find that all ALIGN-KGC variants outperform KGCUNION baseline by vast margins. We notice that asymmetric models perform better for more frequent relations. Because a much larger fraction of triples (see Figure 2) expresses these highly frequent relations, asymmetric models improve aggregate KGC performance. Because RA has not been studied on DBP5L to our knowledge, these are the best numbers yet.

We also note that Jaccard and Asymmetric models do not have any learnable parameters to explicitly encourage EA — any EA gains are due to better RA and KGC. Since those models improve EA performance substantially, we have evidence that other tasks can in turn aid better EA. The learnable asymmetric model, of course, explicitly incentivizes better alignment, leading to a further improvement

Simply computing mBERT embeddings of entity alias texts beats all ALIGNKGC variations that do not use mBERT, by a large margin. This is corroborated by Tang et al. [2020]: mBERT-based EA beats most other baselines reviewed in Section 5. However, joint training with the combination of mBERT with SoftAsymmetric shows a very significant improvement in EA, over and above mBERT. A more detailed report is in Appendix G.

### 4.4 Effect of the number of seed alignments

We also study the effect of the number of seed alignments available to the model at train time, on task accuracies. For this we vary the percentage of seed relation alignments (RA%) and entity alignments (EA%) on x and y axes, and observe KGC and EA HITS@1 performances. Because of diverse and unknown frequencies in its pretraining corpus, the entity alias embeddings provided by mBERT may be of variable quality. To better control for the amount of information available to the model as we vary EA%, we use the SoftAsymmetric model for this experiment.

Figure 5(a) shows KGC hits@1 and Figure 5(b) shows EA hits@1 against EA% and RA%. We observe that (predictably) increasing both EA% and RA% improves KGC performance. At any RA% value, increase in EA% has a powerful impact on both KGC and EA performances. Owing to the collapsing of entity nodes, it increases the number of triples that are seen in some language — these are easier to reason with for ALIGNKGC. Moreover, the model is able to infer many relation IDs as synonymous when more EAs are known, due to the explicit incentivization via asymmetric loss in the loss function — these significantly improve downstream KGC performance.

At low EA%, increasing RA% has limited KGC impact. In this setting, entities in different languages have limited connections, and so the KGs are sparsely connected. Increasing RA% adds further information on fact similarity across KGs, thus improving EA score, but the numbers remain low, due to the overall difficulty of the task.

At high EA% values, increasing RA% does not further improve EA performance. This is because aligned entities (as in the DBP5L data) generally target more frequent relations. Even at low RA%, this enables adequate alignment of frequent relations, leading further to adequate alignment of the entities attached to such relations. However, we posit that it still improves alignment of other entities (not in our gold set), leading to a substantial increase in KGC performance.

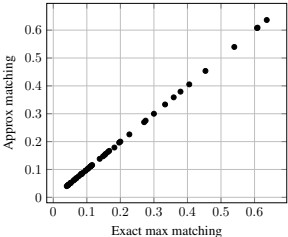

Figure 6: Our approximation for soft overlap is accurate.

### 4.5 Soft overlap approximation quality

We described in Section 3.2.3 our fast approximation for maximal matching, required to evaluate the overlap between $SO(r_1)$ and $SO(r_2)$. In Figure 6 we show a scatter plot of a sample of relation pairs; the x-axis is the true optimal overlap found via the Hungarian matching algorithm, and the y-axis is our approximation. We can see that the approximation is very close to the optimal.

## 5. Related work

**KG Completion:** KGC, through learning embeddings for entities and relations, is a densely-populated research landscape. Among the best performers are ComplEx [Trouillon et al., 2016], ConvE [Dettmers et al., 2018], and RotatE [Sun et al., 2019]. Almost all such systems are designed for a single KG, or are agnostic to the language used in entity and relation aliases.

**KG alignment:** Interest in KG alignment has grown rapidly of late. MTransE [Chen et al., 2017] uses TransE [Bordes et al., 2013] on each KG, while penalizing large distances between embeddings of equivalent entities. BootEA [Sun et al., 2018] is semi-supervised bootstrap EA algorithm. JAPE [Sun et al., 2017] builds attribute- and network neighborhood-based embeddings of entities and combines them with EA constraints. MuGNN [Cao et al., 2019] uses a graph neural network (GNN) to find node embeddings, which are compared to propose ≡ links. AliNet [Sun et al., 2020] also uses a GNN, with an attention mechanism over larger node neighborhoods. BERT-INT [Tang et al., 2020] combines local mBERT features with 1-hop neighborhood interactions. MultiKE [Zhang et al., 2019] treats EA and RA at par. However, they use relation names rather than their SO-signatures. JEANS [Chen et al., 2021] links KG entities to a text corpus. It uses TransE for the KG, skip-grams for the text, with additional alignment constraints.

## 6. Conclusion

We have presented ALIGNKGC, a system that jointly learns to complete multiple monolingual KGs in different languages (KGC) and align their entities and relations. KGC and entity alignment are known tasks, but relation alignment, to our knowledge, has never been integrated with them. ALIGNKGC operates on the KG constructed by taking a union of all monolingual KGs, and extends KGC models to use novel EA and RA loss terms. In extensive experiments, ALIGNKGC significantly improves KGC accuracy, as well as alignment accuracy, underscoring the value of joint alignment and completion.

**Acknowledgments:** Soumen Chakrabarti is partly supported by a Jagadish Bose Fellowship, grants from Google and IBM, and a Halepete Family Chair. Mausam is supported by grants from Huawei, IBM, Google, Bloomberg and 1MG, a Visvesvaraya faculty award by Govt. of India, and Jai Gupta Chair professorship. We thank the IITD HPC facility for compute resources.

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

# Multilingual Knowledge Graph Completion
# With Joint Relation and Entity Alignment
# (Appendix)

## Appendix A. Example of KGC, EA and RA synergy

### A.1  EA helps RA

Suppose a KGC+alignment system is given the equivalences between English (En) and Greek (El) entity names:

- Botafogo_de_Futebol_e_Regatas ≡ Μποταφόγκο_ντε_Φουτεμπόλ_ε_Ρεγκάτας (a football club)
- Brazil ≡ Βραζιλία

Now suppose the system sees these two triples in En and El KGs, respectively:

**En:** (Botafogo_de_Futebol_e_Regatas, home_arena, Brazil)

**El:** (Μποταφόγκο_ντε_Φουτεμπόλ_ε_Ρεγκάτας, σπίτι αρένα, Βραζιλία)

If this pattern occurs often enough, the system can infer the relation alignment "home_arena" = "σπίτι αρένα". This is an example of inferring RA from EA.

### A.2  EA, RA helps KGC

Suppose the system knows that

- Rot-WeissEssen (English) ≡ Ροτ_Βάις_Έσσεν (Greek) (a football club)
- Essen (English) ≡ Έσσεν (a town in Germany)
- (Rot-WeissEssen, homeArena, Essen) holds in the English KG

Then the system can potentially infer (Ροτ_Βάις_Έσσεν, σπίτι αρένα, Έσσεν) in the Greek KG. This is an example of EA and RA helping KGC.

### A.3  KGC helps EA

Now suppose the KGC system has already learnt the (soft) inference pattern

> ([club], home_arena, [city]) and
> ([city], country, [country])
> $\implies$ ([club], home_arena, [country])

where home_arena and country (Χώρα in Greek) are assumed to be 'universal' across languages.

Now if we know that (Έσσεν, Χώρα, Γερμανία) holds in the Greek KG, we can apply the above rule to infer (Ροτ_Βάις_Έσσεν, σπίτι αρένα, Γερμανία).

If the corresponding fact was already in the English KG as (Rot-WeissEssen, homeArena, Germany), Then we can infer the EA Γερμανία ≡ Germany.

## Appendix B. Differentiable set representations

Given a finite set $X = \{x_i\}$ where $x_i \in \mathcal{X}$, Zaheer et al. [2017] showed that any set function $f(X)$ that is invariant to permutations of $X$ can be expressed as $\rho\left(\sum_{x \in X} \phi(x)\right)$ for suitable transformations

| Model_NAME | mBERT | | mBERT(comb) | | AlignKGC | | AlignKGC+mBERT(comb) | |
|---|---|---|---|---|---|---|---|---|
| Language Pair | HITS@1 | HITS@10 | HITS@1 | HITS@10 | HITS@1 | HITS@10 | HITS@1 | HITS@10 |
| EL-EN | 0.51383 | 0.72614 | 0.72257 | 0.88492 | 0.40370 | 0.56633 | **0.82596** | *0.89871* |
| EN-EL | 0.595 | 0.80107 | 0.78234 | *0.91258* | 0.46077 | 0.60627 | **0.83594** | 0.91012 |
| EL-ES | 0.58333 | 0.77564 | 0.76465 | 0.91026 | 0.50606 | 0.66363 | **0.88787** | *0.94848* |
| ES-EL | 0.63828 | 0.81319 | 0.80769 | 0.92766 | 0.53333 | 0.70454 | **0.89393** | *0.95151* |
| EL-FR | 0.48316 | 0.70124 | 0.71099 | 0.86259 | 0.51510 | 0.66767 | **0.86555** | *0.92749* |
| FR-EL | 0.57358 | 0.78014 | 0.75443 | 0.88475 | 0.53776 | 0.69335 | **0.87009** | *0.93504* |
| EL-JA | 0.39668 | 0.60609 | 0.65683 | 0.82196 | 0.53420 | 0.70358 | **0.84364** | *0.91856* |
| JA-EL | 0.44465 | 0.65314 | 0.70111 | 0.8441 | 0.57980 | 0.74592 | **0.85993** | *0.93159* |
| JA-EN | 0.42445 | 0.67555 | 0.71135 | 0.87293 | 0.40239 | 0.54749 | **0.81077** | *0.88332* |
| EN-JA | 0.44279 | 0.69345 | 0.72707 | 0.87686 | 0.41436 | 0.56320 | **0.81899** | *0.88406* |
| JA-ES | 0.41312 | 0.66865 | 0.70443 | 0.85775 | 0.48200 | 0.64765 | **0.83910** | *0.91853* |
| ES-JA | 0.44591 | 0.68101 | 0.70613 | 0.86627 | 0.47318 | 0.64426 | **0.84046** | *0.91242* |
| JA-FR | 0.38099 | 0.64998 | 0.69962 | 0.85617 | 0.53307 | 0.69428 | **0.84906** | *0.91971* |
| FR-JA | 0.40899 | 0.67204 | 0.71022 | 0.85829 | 0.51123 | 0.68143 | **0.84585** | *0.91393* |
| ES-FR | 0.85746 | 0.91545 | 0.89951 | 0.95042 | 0.49034 | 0.64735 | **0.91152** | *0.96697* |
| FR-ES | 0.86012 | 0.91589 | 0.89907 | 0.95263 | 0.49968 | 0.65482 | **0.91962** | *0.97071* |
| ES-EN | 0.92461 | 0.96577 | **0.93562** | *0.98207* | 0.40906 | 0.57358 | 0.91326 | 0.96810 |
| EN-ES | 0.91891 | 0.96373 | **0.93562** | *0.9784* | 0.42137 | 0.59149 | 0.92893 | 0.97425 |
| EN-FR | 0.86634 | 0.92003 | **0.89604** | *0.95211* | 0.36602 | 0.51616 | 0.88383 | 0.95178 |
| FR-EN | 0.86558 | 0.92003 | **0.8968** | *0.95316* | 0.34246 | 0.50246 | 0.866301 | 0.94027 |

Table 7: EA on DBP5L: detailed performance on all language pairs. Best HITS@1 in **bold**. Best HITS@10 in *italics*.

$\rho$ and $\phi$. In practice, capturing correlations between features of $x$s requires unreasonably deep/wide networks for $\rho$ and $\phi$. Pabbaraju and Jain [2019] proposed to present $X$, after applying adversarial permutations, to an order-sensitive recurrent network which can better capture correlations among $x$s, while using lower network capacity. The RNN must reduce variability of its set encoding in the face of these permutations. Lee et al. [2019] replaced the RNN with a transformer network.

## Appendix C. Matching via matrix scaling

Consider two sets $\{x_i : i \in [I]\}$ and $\{y_j : j \in [J]\}$ with a metric $d_{ij} = d(x_i, y_j)$. Let $T_{ij} \in \mathbb{R}_+$ be a *transportation matrix*, subject marginal constraints to $\sum_i T_{ij} = c_j$ for all $j$ and $\sum_j T_{ij} = r_i$ for all $i$. Often, $r_i = c_j = 1$ or $r_i = 1/I$ and $c_j = 1/J$. Our goal is to $\min_T \sum_{i,j} d_{ij} T_{ij}$. While this can be solved via linear programming, it is not clear how to backpropagate losses based on the optimal transportation back to networks that output $\{x_i\}, \{y_j\}$. (In our case, these set elements are functions of KG embedding vectors, through SO-sets.) Cuturi [2013] showed that if the objective is mildly augmented with an entropy term $\lambda H(T) = -\lambda \sum_{i,j} T_{ij} \log T_{ij}$, then the optimal transport can be approximated by iterative row and column scaling of $(d_{ij})$, which are differentiable operations. However, this is computationally expensive for large KGs.

## Appendix D. Exploiting relation descriptions

Along with entities, relations, too, come with textual aliases in some KGs like WikiData. If we denote the textual description of relation $r$ using $t(r)$, we can also use $\mathrm{mlp}(\mathrm{mbert}(t(r)))$ as additional intrinsic features of $r$. (This MLP is different from the one for entities.) Similar to entity alignment, if $b(r_l \Leftrightarrow r_{l'})$ is high, then we want $\mathrm{mbert}(r_l) \approx \mathrm{mbert}(r_{l'})$. Accordingly, we can assess another

loss term $L_{\mathrm{RA3}} =$

$$\sum_{r_l \equiv r_{l'}} b(r_l \Leftrightarrow r_{l'}) \left\| \mathrm{mbert}(r_l) - \mathrm{mbert}(r_{l'}) \right\|_1. \tag{9}$$

In ongoing work, we are preparing a new data set based on WikiData to evaluate the above extension.

## Appendix E. Soft SO-overlap generalizes discrete SO-overlap

**Proposition 1.** If entities are represented as 1-hot vectors (i.e., not using a continuous embedding space), then soft overlap in Definition 4 reduces to discrete overlap in Definition 2.

Proof sketch: it is easy to verify that with 1-hot vectors, soft overlap is exactly the size of the intersection of the two SO sets.

## Appendix F. Other ways to use mBERT signals

Beyond what is presented in the main paper, we explored some other ways to incorporate mBERT outputs, but these did not perform as well as the approach we chose in the end. One idea was to augment the soft SO signature with vectors output from BERT. Specifically, we redefine $\mathrm{SO}(r)$ as

$$\left\{ \left( \boldsymbol{s}, \mathrm{mbert}(s) \right); \boldsymbol{o}, \mathrm{mbert}(o) \right) : (s, r, o) \in \mathrm{KG} \right\} \tag{10}$$

This MLP may be trained as BERT is pinned down, or BERT may also be fine tuned. This suggests the alternative loss term

$$L_{\mathrm{EA}} = \sum_{e_l \equiv e_{l'}} \left\| \mathrm{mbert}(e_l) - \mathrm{mbert}(e_{l'}) \right\|_1 \tag{11}$$

We may not want a flat cosine (as matrix $A_{r_l, r_{l'}}$ was defined), but introduce some and-like semantics for the match between the vectors with four fields $(s_1, s_1'; o_1, o_1')$ and $(s_2, s_2'; o_2, o_2')$. As one example, we might use

$$\sigma\left( \cos((s_1, s_1'), (s_2, s_2')) \right) \, \sigma\left( \cos((o_1, o_1'), (o_2, o_2')) \right). \tag{12}$$

Relative weights balancing $\boldsymbol{e}$ and $\mathrm{mbert}(e)$ may also help.

## Appendix G. Detailed EA evaluation for all language pairs

In Table 7 we show EA performance on all pairs of languages, with one as source and other as target. For the two 'mBERT' columns, we fine-tune 10 mBERT models, one for each of $\binom{5}{2}$ language pairs. For the two 'mBERT(comb)' columns, we combine all the language entity pairs to train a single global mBERT model, which gives a benefit on all language pairs. Adding mBERT(comb) to AlignKGC provides structural information to improve EA performance significantly in a majority of language pairs, particularly the ones with smaller, sparser KGs.

## Appendix H. Standard deviation over random initialization

The table below shows the standard deviation over three runs corresponding to AlignKGC+mBERT. Usually it is considerably smaller than 1%.

| | El | | | Ja | | | Fr | | | Es | | | En | | |
|---|---|---|---|---|---|---|---|---|---|---|---|---|---|---|---|
| | H@1 | H@10 | MRR | H@1 | H@10 | MRR | H@1 | H@10 | MRR | H@1 | H@10 | MRR | H@1 | H@10 | MRR |
| | 1.01 | 0.261 | 0.607 | 2.826 | 0.685 | 1.872 | 0.632 | 0.225 | 0.538 | 1.07 | 0.118 | 0.785 | 0.815 | 0.744 | 0.876 |

# Appendix I. Statistical significance

The table below shows the $p$-values for a t-test between KGCUNION and Jaccard.

| Greek | | | Japanese | | | French | | | Spanish | | | English | | |
|---|---|---|---|---|---|---|---|---|---|---|---|---|---|---|
| H@1 | H@10 | MRR | H@1 | H@10 | MRR | H@1 | H@10 | MRR | H@1 | H@10 | MRR | H@1 | H@10 | MRR |
| 1.03E-07 | 4.1E-07 | 2.07E-08 | 2.84E-05 | 1.09E-06 | 2.85E-06 | 3.3E-06 | 1.23E-07 | 2.47E-07 | 6.19E-09 | 1.76E-08 | 7.93E-10 | 9.23E-06 | 2.28E-04 | 1.18E-05 |

# Appendix J. Code and data

Please visit https://www.cse.iitb.ac.in/~soumen/doc/AlignKGC/

