# OpenReview forum: "Multilingual Knowledge Graph Completion With Joint Relation and Entity Alignment"
_AKBC.ws/2021/Conference — AKBC 2021_

### Official Review · Reviewer_jXN7 · 2021-07-21
**Well-motivated question and well-written, but baselines lacking**

**Rating:** 7
**Confidence:** 3

**Review:**

This paper explores how entity alignment (EA), relation alignment (RA), and knowledge graph completion (KGC) can benefit each other. It proposes a model which jointly optimizes the three tasks using a linear combination of their losses. A contribution within this is a method for aligning relations in the absence of gold entity alignments. KGC is evaluated on each of 5 languages for which a KG is available, and EA & RA are evaluated jointly. The paper is easy to follow throughout.

The results indeed show that information from each of the tasks helps the other two, positively answering the proposed question. There is no baseline which tests the hypothesis in the simplest way, which would be to separately perform EA/RA before KGC. This more powerful baseline would motivate the proposed system more strongly.

- Strengths
    - The methodology section is clear and the reasoning leading to the soft overlap method is easy to understand. The idea of representing a relation as a bag of (s, o) pairs is simple but powerful.
    - The analysis in §4 is strong. Table 3 is appropriately split for facts which are present in a KG for another language than the target.
    - The results demonstrate that combining the three tasks substantially improves performance.
    - Code and data are already released.
- Weaknesses
    - §3.2 identifies that comparing relation embeddings is a reasonable method for relation alignment, but states that this is insufficient and that the proposed approach is necessary. However, I cannot identify an experiment which backs this up (for example by replacing L_{RA-SA} with a loss that only compares relation embeddings).
    - The baselines are insufficient to prove that jointly modelling the three tasks improves performance over a sequential approach, despite the well-formulated argument as to why this is likely to be the case. A suitable addition would be perform EA as a step in the existing KGCunion baseline, which would make for a fairer comparison with the proposed method.
    - Contributions around the role of RA in this task are diminished somewhat by the fact that the task is artificially created, and there exists a 1-1 relationship between relations across languages.
- Question
    - In §4.4/figure 5, why are 4 levels of EA % used when evaluating KGC and only 3 for EA?
- Other edits
    - In instances where results are described as significant (e.g. §4.2 "Jaccard significantly helps"; §4.3 "very significant improvement in EA"), please clarify the details of significance testing carried out
    - Figure 5 - the 3d plots are slightly difficult to understand. They would be easier to understand as line graphs with one of the independent variables represented by a different color.
    - §4.1 actually closer to 70% than 60% of relation labels only have descriptions in one language.
    - §4.1 "However, these relation labels account for only 8% of fact triples" may be more clearly rewritten as only 8% of triples use a relation with a label. A similar change would help make the next sentence easier to understand too.
    - §4.2 "test facts are part of the unseen test set, Qualitatively,": first comma should be a period.
    - §4.3 "explicitly encourage EA —- any": there are two dashes
    - §A.3 "Now if we know that ( ́Εσσεν), Χώρα": remove extra ")"
    - Some references are missing the full author list (e.g. 'Bootstrapping entity alignment with knowledge graph embedding').
    - Some references cite an arXiv version where the published version may be more appropriate.
        - "Cross-lingual Entity Alignment with Incidental Supervision" EACL'21
        - "Learning Latent Permutations with Gumbel-Sinkhorn Networks" ICLR'18
        - "Cross-lingual Entity Alignment via Joint Attribute-Preserving Embedding" ISWC'17
        - "RotatE: Knowledge Graph Embedding by Relational Rotation in Complex Space" ICLR'19
        - "Deep Sets" NIPS'17

---

> ### Author Response · Authors · 2021-07-30
> **Discussion on baselines and data sets**
>
> Thanks for the constructive review.
>
> Weakness 1: Sorry, our comment in Section 3.2 was not worded right. In fact, to compare two relations, we indeed compare their embeddings (via cosine distance -- as mentioned at bottom of Page 6). However, when training these embeddings, we found that a simple union of facts did not yield the best results (KGCUnion). Instead, guiding the training through an auxiliary loss, computed using \emph{relation signatures} helped the model learn better embeddings.  We have rewritten the said text in the paper.
>
> We are not entirely sure what you mean by replacing L_{RA-SA} with a loss that only compares relation embeddings. Please note that this loss is computed only over “silver” aligned relation pairs, which are themselves computed based on SO overlap. So, in some sense the use of relation signatures is essential for this auxiliary loss. Did you have an alternative in mind? For example, doing all pair relation embedding differences will only confuse the model.
>
> Weakness 2 (“...perform EA as a step in the existing KGCunion baseline, which would make for a fairer comparison”): Your comment implies that EA has not been incorporated into KGCUnion. That is not entirely accurate -- since any initial EA supervision is incorporated in KGCUnion (and all other models) by collapsing all pairs of (known) aligned nodes into single nodes. The KGC run can, in principle, extrapolate this information and infer other entity/relation alignments through learned embeddings.
>
> Maybe what you meant was to incorporate “mBERT-based” EA into KGCUnion? If that is the case, please see the response of Weakness 3 by reviewer K7mZ.
>
> Weakness 3 (Role of RA somewhat diminished because in DBP-5L the relation catalog is universal and without relation aliases in multiple languages): We accept this criticism. We are in the process of putting together a more challenging benchmark starting from WikiData. That said, we note that this is the first work that studies the three tasks jointly.
>
> Question: In §4.4/figure 5, why are 4 levels of EA % used when evaluating KGC and only 3 for EA?  --- Because the last level uses 100%EA in training and leaves none for testing, so EA evaluation is not possible.
>
> Significance claims vs. tests (e.g. §4.2 "Jaccard significantly helps"; §4.3 "very significant improvement in EA") --- We performed paired t-tests for Table 3, and found that Jaccard is significantly better than KBCUnion with very small p-values.  See Appendix I in the updated manuscript. More comprehensive significance tests will be reported in the final version.
>
> 3d plots → color and/or 2d --- agreed, fixed.
>
> §4.1 "However, these relation labels account for only 8% of fact triples" (etc., rewrite) --- ”However, relations with monolingual aliases account for only 8\% of fact triples.  Meanwhile, almost 80\% of fact triples use a relation label that has aliases in all five languages.”  (modified in the paper)
>
> Missing conferences in citations --- fixed, thanks.

---

### Official Review · Reviewer_K7mZ · 2021-07-21
**Extensive experimentation, nice approach, strong results; A few questions left unanswered**

**Rating:** 7
**Confidence:** 4

**Review:**

This paper introduces AlignKGC, a neural model that performs knowledge graph completion (KGC) on multilingual knowledge graphs that have entity and relation overlap. AlignKGC makes use of some of this overlap during training to optimize its parameters for two auxiliary tasks, entity alignment (EA) and relation alignment (RA). The resulting model demonstrates strong empirical results on DBP5L for all three tasks, and the paper demonstrates through extensive experiments the efficacy of each model components introduced.

**Strengths**

1. The paper is well-structured and relatively easy to follow. The authors motivate different approaches for relation alignment (which involves pairs of entities in different monolingual KGs, thus less straightforward compared to entity alignment), and progressively introduce more expressive variants with intuitive examples. The proposed model, AlignKGC, is reasonably well-motivated.

2. The proposed AlignKGC model achieves strong empirical results on all five languages in DBP5L, and the paper presents extensive analysis to demonstrate the effect of each model component, that of cross-lingual transfer, and that of cross-task transfer.

**Weaknesses**

1. As one of the main components, Asymmetric relation alignment was motivated by introducing relations that might have entailment relationships between them. Later in Section 4.1, the paper instead states that "Note that no asymmetric gold relation implication r1 ⇒ r2 are available, only equivalences r1 ≡ r2." Why is Asymmetric RA empirically helpful on this dataset? The authors also mention in passing that Asymmetric is worse than Jaccard on the unseen test set, but the provided reason "removal of some reasonable inferences" does not seem directly connected to this motivation.

2. Lack of variance information for performance numbers. While the gap for main results are clearer, it is not difficult to observe performance fluctuation in Figure 5 as %RA and %EA increase monotonically. How much of the performance variation can be attributed to random variation vs actually from more seed entity/relation pairs? More careful analysis is left desired.

3. mBERT is introduced as the last component to enhance entity alignment. Given the observed correlation between the tasks in the analysis (e.g., Figure 5), it remains unclear if some of the improvements brought by different RA approaches will be rendered obsolete (or at least made less pronounced) if mBERT-based EA were introduced first.

4. Very limited comparison to other baseline models. The paper compares to virtually only one baseline KEnS$_b$, aside from author-implemented variants of their own model. Could the authors have compared to baselines mentioned in the related work section, e.g., MultiKE which shares similar motivations (and just on EA/RA tasks on DBP5L?).

**Minor questions to the authors**

1. What exactly is the form of SoftOv, and why is the additional sigmoid over $A_{r_l, r_{l'}}[i,j]$ necessary?

2. In the second-to-last paragraph on Page 8, the paper discusses "with high negative sampling" despite never mentioning negative sampling involved. Could the authors elaborate?

**Clarification after author response**
Thanks for your response!

Just to clarify my point on W3: I agree that testing the contribution of different RA formalization without mBERT as a confounder is valid, and the results that are already presented in the paper are valuable. I was hoping that the authors could demonstrate whether mBERT has already "picked up" enough information about RA to shrink or even eliminate the difference between these. I think this observation will be helpful regardless of the outcome for future work in this direction.

---

> ### Author Response · Authors · 2021-07-30
> **Asymmetric similarity; variance measurements; mBERT discussion**
>
> Thanks for the constructive review.
>
> Weakness 1:  A problem with Jaccard similarity is that it can give a high score to non-synonymous relation pairs that actually have an asymmetric implication between them. E.g., in the DBP5L data set, Jaccard similarity gives a large symmetric score between locationCity and headquarter, or keyPerson and founders relations in different KGs. However, these relation pairs clearly involve asymmetric implication, and should not be linked. Asymmetric computation of r=>r’ (eqn 4) allows the model to guess those relation pairs that may have asymmetric implications (even without any gold data being present for relation implications). The loss term in eqn (6) does not incentivize the model to bring those relation pairs together, thus allowing the model to maintain separate embeddings for such relation pairs. This avoids some erroneous inferences,  generally boosting precision --- possibly at the expense of some recall, as this may remove some “reasonable inferences”as well.
>
> Weakness 2: We will show confidence intervals for the final version in Figure 5. For now, we have computed the standard deviation of our main model (Asymmetric+mBERT) across 3 runs. Please see Appendix H in the revised paper.
>
> Weakness 3: Your point about giving mBERT support to other variations/ablations of AlignKGC rather than at the very end is a valid one. However, as Reviewer 8DZf has noted, mBERT has likely already picked up (and rote-learnt) relational information from its training corpus. mBERT, thus redundantly informed, may blur the finer distinctions between Jaccard, asymmetric and soft asymmetric, adding a “noise floor” to our observations, but we are not sure what insights that would lend. In fact, this might be a reason many EA papers (below) do not cite or compare against BERT-INT.
> https://arxiv.org/pdf/2106.03619.pdf
> https://arxiv.org/pdf/2005.05607.pdf
> https://arxiv.org/pdf/2003.07743.pdf
>
> Weakness 4 (Other EA baselines beyond mBERT, BERT-INT, and KEnSb): Most EA baselines are applied between strictly two languages. Our work addresses the more realistic setting where one target language benefits from simultaneous alignment to a union of multiple ‘source’ language KGs. We are in the process of adapting such EA systems to our setting to get comparable numbers, and expect to include them in the final version.
>
> Q1 (Exact form of SoftOV):
> Recall that $A$ is the matrix of cosine similarities. From the all-to-all possibilities, we thin down $A$ by retaining only those $[i,j]$ pairs where $i$ prefers $j$ most and vice versa. We essentially want to count the number of edges in this restricted matching, but in a way that we can back-propagate the task losses. A soft form of this counting is the sigmoid applied on the scaled and shifted cosine similarity. That is why SoftOv is not the `hard’ number of edges in the matching, but the sum of the sigmoids over the best-partner edges. (Strictly speaking, the loss based on SoftOv is not smooth. This can be addressed via Gumbel-Sinkhorn matrix scaling, but we did not need that complication in practice.)
>
> Q2 Here we are talking about the negative sampling used in the ComplEx part of our loss function, where we used 2,000 negative samples for each positive (s,r,o) triple.

---

### Official Review · Reviewer_8DZf · 2021-07-22
**Interesting approach combining KBC with entity and relation alignment**

**Rating:** 7
**Confidence:** 4

**Review:**

Authors propose AlignKGC, a system for jointly learning to complete multiple monolingual KGs, and aligning their entities and relations. Given a set of multi-lingual KGs, AlignKGC introduces entity alignment and relation alignment loss terms in standard KBC models.
In downstream KBC experiments, the proposed model significantly improves the link prediction accuracy, as well as the accuracy of entity and relation alignment metrics.

The paper is extremely clear, and results on DBP5L show significant improvements over the baselines.
Question: is it possible to use, as a baseline alternative to mBERT, a neural model that does not rely on transfer learning? My main concern with using large models pre-trained on large quantities of data for KBC tasks is that they might have been pre-trained on
textual information that overlaps with the test sets at hand.

Typos/minor issues:
- Eq. 4 introduces an operator, but the operator being used in Eq. 6 seems different -- is it a typo?
- Missing period before "Qualitatively".
- Is it possible to make the plots in Fig. 5 2D? Right now, they are a bit hard to parse.

---

> ### Author Response · Authors · 2021-07-30
> **Baseline discussion; minor typos/notation issues**
>
> Thanks for the constructive review.
>
> EA baselines beyond mBERT: Yes, your concern is valid: we don’t know how much of various KGs mBERT has already effectively seen in its training corpus. At the same time, mBERT performance has exceeded recent non-textual EA approaches. Because we wished to show the value of joint training, even on a very powerful model, we chose mBERT as the baseline. We anticipate that for other weaker models, the performance gain will be even higher, even if the raw numbers will be lower.
>
> “Eq. 4 introduces an operator, but the operator being used in Eq. 6 seems different”
> Perhaps you are pointing out that eqn (4) defines $b_A(r_l \Rightarrow r_{l’})$, whereas eqn (6) uses $b_A(r_l \Leftrightarrow r_{l’})$? Note that the latter is defined just below eqn (4) in running text, as the minimum of the two asymmetric beliefs.
>
> Yes, we have converted Fig 5 into 2d line charts; please see the updated manuscript. Our observations remain unaffected.

---

### Decision · Program_Chairs · 2021-08-17

**Decision:**

Accept

**Comment:**

This paper introduces AlignKGC, a neural model that performs entity alignment (EA), relation alignment (RA), and knowledge graph completion (KGC) on multilingual knowledge graphs. The model jointly optimizes for the linear combination of losses for all 3 tasks. The authors present a convincing set of experiments to show that each of these tasks helps the other two. The paper is well written and demonstrates strong empirical results on DBLP5L containing 5 languages on all 3 tasks. Results are reproducible as both the code and data are made public. No major concerns after the author's response.